# MCT-Dependent *Cryptosporidium parvum*-Induced Bovine Monocyte Extracellular Traps (METs) under Physioxia

**DOI:** 10.3390/biology12070961

**Published:** 2023-07-05

**Authors:** Seyed Sajjad Hasheminasab, Iván Conejeros, Ulrich Gärtner, Faustin Kamena, Anja Taubert, Carlos R. Hermosilla

**Affiliations:** 1Institute of Parasitology, Biomedical Research Center Seltersberg (BFS), Justus Liebig University Giessen, 35392 Giessen, Germany; ivan.conejeros@vetmed.uni-giessen.de (I.C.); anja.taubert@vetmed.uni-giessen.de (A.T.); carlos.r.hermosilla@vetmed.uni-giessen.de (C.R.H.); 2Institute of Anatomy and Cell Biology, Justus Liebig University Giessen, 35392 Giessen, Germany; ulrich.gaertner@anatomie.med.uni-giessen.de; 3Laboratory for Molecular Parasitology, Department of Microbiology and Parasitology, University of Buea, Buea P.O. Box 63, Cameroon; faustinkamena@googlemail.com

**Keywords:** *Cryptosporidium parvum*, METosis, monocytes, Notch, purinergic receptor P2X1

## Abstract

**Simple Summary:**

*Cryptosporidium parvum* is a zoonotic apicomplexan parasite causing severe enteritis with high morbidity and mortality in young children, immunosuppressed patients, and newborn calves. Monocytes are bone marrow-derived myeloid leukocytes with a central role in early host innate immunity to infections and injury. Monocytes sense invasive pathogens, phagocytose, secrete cytokines/chemokines, present antigens to T cells, and release monocyte extracellular traps (METs). This study aimed to investigate the involvement of ATP purinergic receptor P2X1, glycolysis, monocarboxylate transporters (MCT), and Notch signaling in METs of bovine monocytes exposed to *C. parvum*. *C. parvum*-induced METosis was investigated under different oxygen conditions, including intestinal physioxia (5% O_2_) and hyperoxia (21% O_2_). *C. parvum*-oocysts, as well as sporozoites, robustly induced METosis and MET-associated anti-parasitic molecules such as myeloperoxidase (MPO) and histones confirmed METs extrusion.

**Abstract:**

The apicomplexan protozoan parasite *Cryptosporidium parvum* is responsible for cryptosporidiosis, which is a zoonotic intestinal illness that affects newborn cattle, wild animals, and people all over the world. Mammalian monocytes are bone marrow-derived myeloid leukocytes with important defense effector functions in early host innate immunity due to their ATP purinergic-, CD14- and CD16-receptors, adhesion, migration and phagocytosis capacities, inflammatory, and anti-parasitic properties. The formation of monocyte extracellular traps (METs) has recently been reported as an additional effector mechanism against apicomplexan parasites. Nonetheless, nothing is known in the literature on METs extrusion neither towards *C. parvum*-oocysts nor sporozoites. Herein, ATP purinergic receptor P2X1, glycolysis, Notch signaling, and lactate monocarboxylate transporters (MCT) were investigated in *C. parvum*-exposed bovine monocytes under intestinal physioxia (5% O_2_) and hyperoxia (21% O_2_; most commonly used hyperoxic laboratory conditions). *C. parvum*-triggered suicidal METs were confirmed by complete rupture of exposed monocytes, co-localization of extracellular DNA with myeloperoxidase (MPO) and histones (H1-H4) via immunofluorescence- and confocal microscopy analyses. *C. parvum*-induced suicidal METs resulted not only in oocyst entrapment but also in hindered sporozoite mobility from oocysts according to scanning electron microscopy (SEM) analyses. Early parasite-induced bovine monocyte activation, accompanied by membrane protrusions toward *C. parvum*-oocysts/sporozoites, was unveiled using live cell 3D-holotomographic microscopy analysis. The administration of NF449, an inhibitor of the ATP purinergic receptor P2X1, to monocytes subjected to varying oxygen concentrations did not yield a noteworthy decrease in *C. parvum*-induced METosis. This suggests that the cell death process is not dependent on P2X1. Additionally, blockage of glycolysis in monocyte through 2-deoxy glucose (2-DG) inhibition reduced *C. parvum*-induced METosis but not significantly. According to monocyte energetic state measurements, *C. parvum*-exposed cells neither increased extracellular acidification rates (ECAR) nor oxygen consumption rates (OCR). Lactate monocarboxylate transporters (MCT) inhibitor (i.e., AR-C 141990) treatments significantly diminished *C. parvum*-mediated METs extrusion under physioxic (5% O_2_) condition. Similarly, treatment with either DAPT or compound E, two selective Notch inhibitors, exhibited no significant suppressive effects on bovine MET production. Overall, for the first time, we demonstrate *C. parvum*-mediated METosis as P2X1-independent but as an MCT-dependent defense mechanism under intestinal physioxia (5% CO_2_) conditions. METs findings suggest anti-cryptosporidial effects through parasite entrapment and inhibition of sporozoite excystation.

## 1. Introduction

The apicomplexan protozoan parasite *Cryptosporidium parvum* is responsible for the zoonotic intestinal illness known as cryptosporidiosis, which affects people, wild animals, and newborn cattle worldwide. *C. parvum* completes its whole life cycle in the small intestinal epithelial cells (IEC) of hosts and significantly contributes to diarrhea morbidity and death in young individuals below five years in developing countries [1,2,3]. When it comes to the duration of *C. parvum* infection and pathogenesis, the immune status of patients seems critical. Immunocompetent persons may experience severe but self-limiting diarrhea, while immunocompromised patients may experience life-threatening and chronic cryptosporidiosis. Within the epizootiology of cryptosporidiosis, highly resistant *C. parvum* oocysts to environmental stressors, including chlorine treatment of community water supplies, seem to play an important role resulting in either water- and food-borne disease transmission, particularly in the developing world [4]. Neonatal calves infected with *C. parvum* also have severe enteritis, which has a substantial negative impact on the global economy. Even more significantly, cattle are considered the most prominent reservoir of *C. parvum* and, thereby, play a crucial role in the epidemiology of cryptosporidiosis in humans [5,6,7].

Immunological investigations on *C. parvum* in vivo have mostly focused on protective responses of the adaptive cellular immune system. It is crucial to remember that the bulk of these conclusions are based on the immune responses of *C. parvum*-infected mice, which have a very different immune system than humans or livestock. This means that mice-derived immune reactions against *C. parvum* may not accurately reflect human or livestock protective cellular immunity in vivo [8,9]. To date, little is known about how monocytes and other related professional phagocytes respond to *C. parvum* during the onset of the disease. Numerous studies on the host’s innate immunity have shown that the establishment of protective immunity against this parasite is largely dependent on monocytes, macrophages, polymorphonuclear neutrophils (PMN), and IEC [9,10,11,12,13,14,15,16,17,18,19,20]. Monocytes are bone marrow-derived myeloid leukocytes with important defense effector functions in early host innate immunity and the cell type to arrive at inflammatory sites of the intestine after PMN recruitment. Monocytes can differentiate into either macrophages or dendritic cells (DCs) in the body before migrating into lymph nodes, depending on where they are positioned. Pathogens, including protozoans and apoptotic cells (e.g., apoptotic PMN), can all be phagocytosed by activated monocytes [21], resulting in the secretion of cytokines/chemokines to recruit more PMN and other leukocytes to the site of infection [21].

Monocytes recognize pathogens via pattern recognition-receptors (PRR), activating defense responses consequently, in response to various pathogens such as bacteria, fungi, protozoa, and viruses [22]. Monocytes are mobilized and present toll-like receptors (TLR), dectin-1 [23], CD18, and CD11b [24] in addition to cytosolic PRR identifying pathogen-associated molecular patterns (PAMP) on their surfaces. In addition, non-classical monocytes are recruited to the site of infection, although in a manner that is not as strong, participating in the primary reaction to inflammation through the secretion of TNFα and various chemokines [22]. Monocytes respond to viral infection or circulatory nucleic acids by means of TLR7/8, damage to endothelial cells using TLR7, and tumor metastasis control through CX3CR1 [25,26]. After PAMP-mediated recognition, monocyte-derived effector mechanisms will occur, including the production of reactive oxygen species (ROS) [27,28], phagocytosis [29,30,31,32,33,34], and extrusion of monocyte extracellular traps (METs) to entrap and kill invasive pathogens [35,36,37,38,39]. The literature reports that certain vaccinations or infections (e.g., measles vaccination or Bacillus Calmette–Guerin (BCG) and *Candida albicans* infection) enhance the long-term reactivity of monocytes, which provides resistance to secondary infections [40,41,42,43,44]. Additionally, METs can play a pivotal role in human reproduction disorders, i.e., epididymitis [37]. In ex vivo seminal fluid samples of epididymitis patients, spontaneous METosis extrusion was detected, resulting in increased spermatozoa entrapment [37]. In line with this finding, in healthy patient-derived seminal fluid smears, reduced sperm mobility due to METs-mediated ensnarement has been demonstrated [37]. Despite all these monocyte findings, METs formation as a novel effector mechanism against protozoan parasites is still very scarce. Exclusively, four reports exist so far on the critical role of METs extrusion against apicomplexan parasites, i.e., *Besnoitia besnoiti* [35], *Toxoplasma gondii* [45], *Neospora caninum* [46] and *Eimeria ninakohlyakimovae* [36]. This is the first study to investigate the effects of *C. parvum* on the generation of METs in bovine monocytes exposed to infective oocysts and sporozoites.

It Is well known that Notch signaling is engaged in a number of distinct signal transduction pathways, which seem essential for the differentiation and homeostasis of immunocompetent cells [47,48]. In line, monocytes, macrophages, and DCs constitutively express receptors and Notch ligands, thus bearing the capacity to induce and respond to the Notch pathway via modulation of TLR [49,50]. We used particular inhibitors in this study to analyze Notch signaling in *C. parvum*-triggered METosis. In addition, we investigated the influence of oxygen circumstances on monocarboxylate transporters (MCT), ATP purinergic P2X1 receptors, and monocyte-derived glycolysis via selective inhibitors.

## 2. Materials and Methods

### 2.1. Ethics Statements

The present study adhered to the guidelines set forth by the Animal Care Committee of JLU Giessen. The Ethic Commission for Experimental Animal Studies of the Federal State of Hesse, as overseen by the Regierungspräsidium Giessen, has granted approval for protocols that adhere to the European Animal Welfare Legislation, specifically ART13TFEU, as well as prevailing German Animal Protection Laws.

### 2.2. Parasite Strain and Excystation of Cryptosporidium parvum Sporozoites

The *C. parvum* strain (IIaA15G2R1) was maintained by multiple passages of oocysts in one-day-old calves at the Institute of Parasitology, University of Leipzig, Leipzig, Germany, as previously reported [51,52]. This strain refers to the *C. parvum* 60-KDa glycoprotein (gp60) as a commonly documented zoonotic subtype of *C. parvum* in advanced economic countries, including Germany [52,53,54,55]. Using a sedimentation/flotation method, *C. parvum* oocysts were isolated [52,56] and maintained in sterile phosphate-buffered saline (PBS; pH 7.4) with 200 μg/mL penicillin/streptomycin and 5 μg/mL amphotericin B (both Sigma-Aldrich, Darmstadt, Germany) at 4 °C. At monthly periods, the medium of *C. parvum* oocyst stock was replenished [52,53]. *C. parvum* oocysts were centrifuged (5000× *g*/5 min/4 °C) and suspended in excystation media to extract sporozoites [52,56]. Sporozoite excystation in *C. parvum* was produced by preheating 1× Hank’s balanced salt solution (HBBS; Sigma-Aldrich, Darmstadt, Germany) to 37 °C and acidifying it to pH 2.0 for 10-30 min. After incubating the sporozoites at 37 °C for 10 min in non-acidic HBSS (Sigma-Aldrich, Darmstadt, Germany), we pelleted the free-floating sporozoites. The sporozoites were suspended in RPMI 1640 cell culture medium with 0.3 g/mL L-glutamine (both Gibco, Darmstadt, Germany), 0.1 mg streptomycin, and 100 UI penicillin (both Sigma-Aldrich, Darmstadt, Germany).

### 2.3. Bovine Monocyte Isolation

Nine healthy adult dairy cattle were used as blood donors. 30 mL of peripheral blood was obtained in heparinized, sterile plastic containers (Kabe Labortechnik, Nümbrecht, Germany). In accordance with Taubert et al. (2009), isolation of peripheral blood mononuclear cells (PBMC) was performed [57]. In sterile 50 mL centrifuge containers (Greiner, Frickenhausen, Germany), 20 mL of heparinized bovine blood was combined with 20 mL of PBS-EDTA and deposited on top of 12 mL of Histopaque^®^ 1077 separating solution (Sigma-Aldrich, Darmstadt, Germany). Following centrifugation at 400× *g* for 45 min, the layer containing PBMC was retrieved and subjected to three washes at 4 °C, each lasting 10 min at 400× *g* using sterile RPMI medium (Gibco, Darmstadt, Germany). Subsequently, PBMC was permitted to attach to 75-cm^2^ tissue plastic flasks (Greiner, Frickenhausen, Germany) that had been coated with 2% sterile gelatine (previously dried after being subjected to a temperature of 37 °C for 2 h) for a duration of 1 h at a temperature of 37 °C and non-adherent PBMC were eliminated by washing with pre-warmed RPMI 1640 including 1% streptomycin/penicillin (Sigma-Aldrich, Darmstadt, Germany). The monocytes were dissociated by incubating them in a solution of 10 mM EDTA in Ca++- and Mg++-free Hank’s solution at room temperature (RT; all Sigma-Aldrich, Darmstadt, Germany) for a duration of 5–10 min. The resulting cells were then subjected to a washing step at 4 °C for 10 min at 400× *g*, followed by resuspension in RPMI 1640 supplemented with 1% penicillin and 1% streptomycin (Sigma-Aldrich, Darmstadt, Germany) and 10% bovine fetal serum (BFS; Sigma-Aldrich, Darmstadt, Germany) while maintaining a temperature of 4 °C.

### 2.4. Extracellular Acidification Rates (ECAR) and Oxygen Consumption Rates (OCR) in Bovine Monocytes Exposed to Cryptosporidium parvum

Following the establishment of baseline measurements, bovine monocytes were subjected to sporozoite exposure, and subsequently, the energy levels of monocytes were measured using a Seahorse XF^®^ analyzer (Agilent, Rathingen, Germany). Monocytes were subjected to centrifugation at 400× *g* for a duration of 10 min (RT). The suspension of the cell pellet was carried out in 0.5 mL of XF^®^ assay media, which was further enriched with 10 mM glucose, 2 mM L-glutamine, and 1 mM pyruvate (all Agilent, Rathingen, Germany). In this experiment, an eight-well XF^®^ Seahorse analyzer plate (Agilent, Rathingen, Germany) was utilized, pre-coated with 0.001% poly-_L_-lysine for 30 min (Sigma-Aldrich, Darmstadt, Germany). 1 × 10^5^ cells were meticulously pipetted into each well. The control utilized in the study was the Plain XF^®^ assay medium. Subsequently, in order to attain a final volume of 180 µL per well, 130 µL of XF^®^ assay media (Agilent, Rathingen, Germany) was introduced into each well, followed by incubation at 37 °C for 45 min without CO_2_ supplementation. During this interim period, a suspension of *C. parvum* (consisting of 3 × 10^5^ oocysts/20 μL) was introduced into XF^®^ assay medium (manufactured by Agilent, Rathingen, Germany), and subsequently injected into one of the designated ports to facilitate exposure of the parasites to bovine monocytes. The determination of the area under the curve (AUC), as well as the assessment of oxygen consumption rates (OCR) and extracellular acidification rates (ECAR), were conducted utilizing Wave^®^ software (Agilent, Rathingen, Germany), as previously documented [58].

### 2.5. Glycolysis, MCT1, MCT2 and ATP Purinergic Receptor P2X1 Inhibition in Cryptosporidium parvum-Triggered METs

In this study, bovine monocytes were subjected to pre-treatment with various inhibitors, including AR-C 155858, AR-C 141990, NF449, and 2-DG, for a duration of 30 min. The purpose of these pre-treatments was to block glycolysis (2-DG), MCT1/2 (AR-C 155858, AR-C 141990), and ATP purinergic receptor P2X1 (NF499), respectively. Following the pre-treatment, the monocytes were co-cultured with *C. parvum* sporozoites at a ratio of 1:3 monocyte/sporozoite for a duration of 3 h, under hyperoxia (21% O_2_) and physioxia (5% O_2_) conditions. Physioxia was selected to mimic the in vivo oxygen pressure of the small intestine, and hyperoxia was the most commonly used oxygen setting of in vitro *C. parvum* investigations.

### 2.6. Notch Signaling Inhibition of Cryptosporidium parvum-Mediated METs under Hyperoxic and Physioxic Circumstances

Freshly isolated monocytes co-cultured with *C. parvum* sporozoites in the ratio of (1:3) in 21% O_2_ (hyperoxia) [52,54,59] and 5% O_2_ (physioxia) circumstances [52,53]. As has already been documented by Vélez et al. [52,53], physioxic conditions were established using an InvivO2^®^ 400 physiological system (Ruskinn, Vienna, Austria), while hyperoxic (21% O_2_) co-culture conditions were established using a standard cell culture incubator (Heracell 240i, Thermo Fischer Scientific, Braunschweig, Germany). 1 × 10^6^ cells/mL (*n* = 5) was in sterile 1× HBSS buffer (Sigma-Aldrich, Darmstadt, Germany) and subjected to two different oxygen conditions. Subsequently, SYTOX Green^®^ (5 µM; Thermo Fisher Scientific, Braunschweig, Germany) was introduced into the sample. A total of 1 × 10^5^ cells were seeded in each well of a transparent plastic microplate, with a volume of 50 μL per well (96-well plate; Greiner, Frickenhausen, Germany). The Notch signaling pathway inhibitors, namely compound E at a concentration of 20 nM and DAPT at a concentration of 50 μM (both obtained from Merck Millipore, Darmstadt, Germany), were introduced. In each experimental condition, a quantity of 3 × 10^5^
*C. parvum* sporozoites was introduced per well. The experiment involved culturing plates under both physioxia and hyperoxia conditions. The extracellular DNA release, which leads to the formation of METs, was tracked over a 120-min period using an automated multi-plate monochrome reader (Varioskan Flash^®^; Thermo Fischer Scientific, Braunschweig, Germany). The fluorescence emitted was measured by the reader at two-minute intervals, utilizing an emission wavelength of 523 nm and an excitation wavelength of 504 nm.

### 2.7. Scanning Electron Microscopy (SEM) Analysis for Cryptosporidium parvum-Induced METs Investigation

Samples were prepared by co-culturing viable *C. parvum* oocysts/sporozoites with bovine monocytes in a 1:3 proportion for a duration of 120 min. The establishment of cultures was carried out on pre-coated coverslips utilizing poly-_L_-lysine 0.01% (Thermo Fisher Scientific, Braunschweig, Germany) and maintained at a temperature of 37 °C and 5% CO_2_ under two different oxygen conditions (i.e., 21% and 5%). Subsequently, the cells underwent fixation using 2.5% glutaraldehyde (Merck Millipore, Darmstadt, Germany) and were subsequently subjected to post-fixation using 1% osmium tetroxide (Merck Millipore, Darmstadt, Germany). Following this, the cells were rinsed with distilled water, dried, treated with CO_2_ to reach the critical point, and ultimately coated with gold particles. The specimens were examined utilizing a scanning electron microscope (Philips XL30^®^, Eindhoven, The Netherlands). To confirm monocyte isolation, we used a commercial mouse anti-bovine CD172a antibody (MCA2041C; Bio-Rad Laboratories, Feldkirchen, Germany).

### 2.8. Visualization of Cryptosporidium parvum-Induced METosis Using Confocal- and Immunofluorescence Microscopy Assays

Bovine monocytes and *C. parvum* sporozoites/oocysts were subjected to co-culture in a ratio of 1:3. The co-culturing process was carried out on sterile glass coverslips with a diameter of 15 mm (Greiner, Frickenhausen, Germany) which were previously treated with 0.01% poly-_L_-lysine (Sigma-Aldrich; Darmstadt, Germany). The co-culturing process was conducted at a temperature of 37 °C and a CO_2_ concentration of 5%, while the oxygen concentration was maintained at 21% and 5% oxygen. The duration of the co-culturing process was 120 min. The cells were fixed using a 4% paraformaldehyde solution obtained from Merck Millipore (Darmstadt, Germany) and subsequently stored at a temperature of 4 °C. In order to disclose components and proteins specific to METs, antibodies against myeloperoxidase (MPO) (1:500; Abcam-ab208670, Darmstadt, Germany) and histones (1:500; Merck-MAB3422, Darmstadt, Germany) were utilized. After three aseptic washes with PBS, a 60-min incubation in 2% bovine serum albumin (BSA) (both Sigma-Aldrich, Darmstadt, Germany) enriched with 0.3% Triton X100 (Thermo Fisher Scientific, Braunschweig, Germany) followed. The samples were then incubated overnight with primary antibodies, as mentioned before. The secondary antibody solutions, IgG #A11008-Alexa 488 goat anti-mouse and IgG #A11005-Alexa 594 goat anti-mouse (Thermo Fisher Scientific, Braunschweig, Germany), were used to incubate the coverslips for a duration of 60 min at RT under complete darkness. Upon completion of the incubation period, the specimens underwent a triple rinse with aseptic phosphate-buffered saline (PBS; Sigma-Aldrich, Darmstadt, Germany). The samples were mounted using an anti-fading buffer containing DAPI (Thermo Fisher Scientific, Braunschweig, Germany) and subsequently rinsed thrice with sterile PBS. In the present study, the observation of *C. parvum*-mediated suicidal METosis was conducted using a Nikon ECLIPSE Ti2 confocal microscope (Melville, NY, USA) and an Olympus IX81^®^ inverted epifluorescence microscope integrated with an XM10^®^ digital camera (Tokyo, Japan). Upon utilizing the ImageJ^®^ software to determine the proportion of cells that constituted METs, it was observed that the parasites had induced the formation of METs. By employing image segmentation techniques, the monocyte nuclei were differentiated from the extracellular traps. The METotic cells were identified as those with decondensed and expanded nuclei and were exclusively considered as monocytes undergoing METs formation.

### 2.9. Investigation of METosis Induced by Cryptosporidium parvum in Live Cells Using 3D Holotomographic Microscopy

In this particular experiment, a sample of bovine monocytes composed of 1 × 10^6^ was subjected to centrifugation at 300× *g* for a duration of 10 min at RT. After discarding the supernatant, the monocytes were resuspended in imaging media comprising 2 μM DRAQ5 and 0.5 μM SYTOX Green^®^ (both Thermo Fisher Scientific, Braunschweig, Germany), along with 0.1% BSA (Sigma-Aldrich, Darmstadt, Germany). Sterilized Ibidi^®^ plates were utilized to seed one ml of the cell solution. The cell solution was subsequently incubated at 5% CO_2_ and 37 °C in an incubation cell chamber (Rovereto, Italy). A 30-min stabilization period was utilized to facilitate monocyte settling and to prevent objective condensation. The dish was inoculated with 3 × 10^6^ sporozoites of *C. parvum* at the center with caution. To capture images for all channels, including SYTOX Green^®^ (green channel), refractive index (RI), and DRAQ5^®^ (red channel), a Fluo-3D Cell Explorer^®^ microscope (Nanolive, Lausanne, Switzerland) was employed at a frequency of one image per minute for a duration of three hours. Upon completion of the experiment, each channel was exported individually. The RI values were utilized at each frame to conduct digital staining via the Steve^®^ program (Nanolive, Lausanne, Switzerland). Additional procedures and analyses were conducted utilizing the Image J^®^ software (Fiji version 1.7; NIH, Bethesda, MD, USA).

### 2.10. Statistical Methods

The criterion for statistical significance was established as a *p*-value of less than 0.05. The *p* values were calculated using GraphPad^®^ Prism software (v.9.1) to perform a Kruskal–Wallis (followed by Dunn’s post-hoc test) for multiple comparisons on the datasets. Each graph depicts the mean value along with the corresponding standard deviation.

## 3. Results

### 3.1. Suicidal METosis Induced by Cryptosporidium parvum-Oocysts and Sporozoites

SEM analyses were conducted to detect METosis triggered by *C. parvum*. The phenomenon of suicidal METosis was verified through the utilization of SEM analysis. The results of the study demonstrated that the exposure of bovine monocytes to *C. parvum* oocysts (as depicted in Figure 1A(A1–C1,E1–G1)) for a duration of 120 min led to the emergence of either thin or thick fibers of chromatin strands. These strands were observed to be released from ruptured monocytes (Figure 1A(A1–C1,E1–G1), black arrows). Moreover, activated monocytes with rough surfaces (Figure 1A(G1), black arrowhead) were also detected in these reactions but without METs extrusion.

Some *C. parvum*-oocysts appeared to be partially covered by extruded METs fibers (Figure 1A(A1), white arrows). Instead, others were nearly entirely coated by suicidal METs (Figure 1A(E1,F1), white arrows). The aforementioned observation also holds true for the sporozoites that were ensnared in monocyte-derived extracellular fibers (Figure 1A(A1,G1)). This finding serves as evidence that the production of METs by *C. parvum* is not a defense mechanism that is specific to a particular stage of the parasite’s life cycle. Additionally, it confirms that bovine METosis can impede both sporozoite excystation and the invasion of host cells by sporozoites, which has been demonstrated in previous studies [60].

The conventional characteristics of mammalian-derived extracellular traps (ETs) in stimulated monocytes induced by *C. parvum* were confirmed through immunofluorescence and confocal microscopy studies utilizing specific antibodies against histones (H1, H2A/H2B, H3, H4) and MPO, along with DNA staining. The co-localization of DNA-positive extracellular fiber backbones with H1-, H2A/B-, H3-, H4-, and MPO-positive signals provides evidence for the occurrence of *C. parvum*-triggered METs (Figure 2). The findings were validated through the utilization of confocal microscopy (as shown in Figure 3) and 3D holotomographic imaging (as depicted in Figure 4) of bovine monocytes during MET formation.

### 3.2. Imaging Cryptosporidium parvum-Mediated METosis Illustration Using Live Cell 3D-Holotomography

In order to enhance the existing findings obtained through immunofluorescence microscopy analysis of fixed cells, a 3D-holotomography study utilizing live cell imaging was conducted to gain deeper insights into the initial interactions between parasites and monocytes, as well as the resulting nuclear morphological changes. The present study employed live cell three-dimensional holotomographic microscopy (3D Cell Explorer^®^, Nanolive, Lausanne, Switzerland) to investigate the initial interactions between bovine monocytes and *C. parvum* oocysts/sporozoites during the METotic process, with the aim of gaining a deeper understanding of the phenomenon. Following a period of 30 min of exposure to either motile sporozoites or oocysts, bovine monocytes underwent activation, as evidenced by the formation of pseudopods and laminopods directed toward these parasite stages. Figure 4C presents illustrations of the interactions between bovine monocytes and parasites, along with Figure 4B, which shows the morphology of non-activated bovine monocytes. These illustrations are accompanied by live cell 3D rendering and RI-based digital staining to enhance their visual clarity. The aforementioned signal exhibits a correlation with alterations in the nuclear configuration of monocytes, ascertained through the utilization of DRAQ5^®^ labeling (red). Figure 4A illustrates the time-lapse experiment. Following a co-culture period of 180 min, an observed increase in SYTOX Green^®^ fluorescence signals (green) was noted. The DRAQ5^®^ staining exhibited a correlation with a modification in the morphology of the nucleus (red). The findings of the study indicate that exposure of monocytes to *C. parvum* resulted in an augmentation of cytoplasmic dimensions and a modification in nuclear configuration. As video number 1 encompasses the complete footage of the experiment.

### 3.3. Monocyte Exposure to Cryptosporidium parvum Neither Affected Oxygen Consumption Rates (OCR) Nor Extracellular Acidification Rates (ECAR)

The metabolic activities of bovine monocytes (*n* = 3) exposed to *C. parvum* were evaluated using specifically the Seahorse XFp^®^ (Agilent, Rathingen, Germany. Cell energy profiles were utilized to measure both OCR and ECAR. Upon obtaining baseline values of ECAR and OCR from plain monocytes, the introduction of live parasites did not yield a prompt and sustained increase in ECAR values. The estimation of AUC did not reveal any enhanced ECAR or enhanced OCR values in monocytes exposed to *C. parvum*, as compared to non-stimulated bovine monocytes (Figure 5).

### 3.4. Cryptosporidium parvum-Mediated METosis Is a Physioxia- and a MCT1/MCT2-Dependent Cell Death Process

Lactate, a significant byproduct of glycolysis, has the potential to cause extracellular acidification upon its release into the extracellular space through the mediation of monocarboxylate transporters (MCT) located in the plasma membrane of bovine monocytes [43]. The potential involvement of MCT1 and MCT2 in the process of bovine METosis induced by *C. parvum* was evaluated. Prior to exposure to *C. parvum*, blockers of MCT1/2 (AR-C 141990) and MCT1 (AR-C 155858) were administered to monocytes. The study found no statistically significant differences between the untreated control group and the group of monocytes treated with AR-C 155858. However, when AR-C 14990 (MCT1/2) was applied, more effects were observed in the group of monocytes treated with AR-C 14190 compared to the untreated control group under both oxygen conditions. The *p*-value for both comparisons was less than 0.05 (Figure 6). The relevance of transporters in *C. parvum*-triggered METosis, particularly in relation to the inhibition of AR-C 14990 under physioxic conditions (5% O_2_), requires further investigation.

In order to provide additional confirmation regarding the significance of glycolysis in the METotic process, bovine monocytes were subjected to pre-treatment with 2-deoxy-D-glucose (2-DG), a glycolysis inhibitor, followed by exposure to parasite stages. Consistent with the aforementioned data, the administered treatments did not significantly decrease the formation of METosis induced by *C. parvum* (in comparison to the non-treated condition). This observation implies that this particular process operates as an effector mechanism independent of glycolysis (Figure 7).

### 3.5. Cryptosporidium parvum-Induced Suicidal METosis Is Neither a P2X1- Nor a Notch-Dependent Cell Death Process

The goal of the current work was to investigate how P2X1 purinergic signaling contributes to the suicidal METosis that *C. parvum* causes. Prior to their introduction into the culture, monocytes were pre-treated with NF449, a particular blocker of the ATP purinergic receptor P2X1, to prevent the ATP cell-signaling pathway. The findings indicate that herein, treatments with NF449 did not exhibit significant inhibition (as shown in Figure 8) of the production of METs induced by *C. parvum*, as compared to the untreated controls (monocytes that were not treated and exposed to *C. parvum* versus monocytes that were treated and exposed to *C. parvum*). Significant differences in METosis induction were detected for bovine monocytes stimulated with *C. parvum* under physioxia (5% O_2_) as well as hyperoxia (21% O_2_) when compared to negative controls (Figure 8).

The Notch-mediated cellular signaling pathway is a highly conserved mechanism of signal transduction that primarily participates in the regulation of homeostasis across various cell types as well as cell differentiation. To date, the involvement of Notch in the process of bovine METosis remains unexplored. Considering that various Notch inhibitors exhibit a preference for distinct host systems [61], in this study, two widely utilized inhibitors, namely compound E and DAPT, were employed to obstruct the receptor’s activating proteolytic processing (blocking gamma-secretase and inhibition of the *N*-receptor cleavage). Following the pre-treatment of bovine monocytes with inhibitors specific to Notch, the quantification of MET release was conducted subsequent to parasite exposure. In general, the two inhibitors resulted in an insignificant decrease in METosis induced by the parasite. The results indicate that compound E had a greater impact on METosis reduction under physioxia (see Figure 9B) compared to Notch treatments with DAPT, which only had a limited effect in reducing MET production under hyperoxia (Figure 9A). In general, these results may suggest a potential involvement of oxygen concentration in the Notch-mediated signaling pathway during *C. parvum*-induced METosis, although additional investigation is required to confirm this hypothesis.

## 4. Discussion

Data on metabolic requirements during apicomplexan-induced METs or NETs are very scarce despite the fact that there is a strong correlation between PMN/monocyte activation and induction of metabolic pathways that have previously been reported in the literature [62,63,64,65,66]. Accordingly, monocytes are capable of shifting from a resting to an activated status, thereby switching metabolic pathways on and off in order to generate energy to fulfill their defense functions [64]. Therefore, activated or resting monocytes can rewire sequentially metabolic as well as bioenergetic demands during acute inflammation responses [64]. The phenotypic and functional changes of monocytes are accompanied by significant alterations in cell metabolism [67,68,69,70,71]. As with PMN, macrophages and monocytes predominantly depend on glycolysis and exhibit two discontinuities in the TCA cycle, leading to the buildup of itaconate, a significant antibacterial agent, and succinate [72].

In this study, the contribution of glycolysis, selective ATP purinergic-, Notch- and MCT-dependent signaling in *C. parvum*-mediated bovine METs against two infective stages, i.e., oocysts and sporozoites, were investigated. Glycolysis-related molecules such as ATP are considered the main source of energy in leukocytes, including monocytes, therefore delivering the energy needed for monocyte-derived effector mechanisms and various cellular functions [65,73,74]. In line with these findings, we hereby investigated the role of 2-DG inhibitors on glycolysis. As a result, we observed that under both oxygen concentrations (5% and 21% O_2_), *C. parvum*-triggered suicidal MET formation was reduced but not significantly.

In the meantime, few studies have so far examined the critical role of METosis as part of early host innate immune responses against protozoan- [35,36,45,46] and macrophage-derived extracellular traps against helminth parasites [75]. Additionally, it has been demonstrated that METosis plays significant roles in human reproduction, as recently evidenced by METs-mediated human spermatozoa aggregates, resulting in reduced sperm motility, previously stimulated with *E. coli* [37] or ex vivo in ejaculates of epididymitis patients where METosis-mediated sperm entrapment was confirmed by confocal microscopy- as well as SEM analyses [37,76]. Another study, this time in female patients, showed that PBMC-purified monocytes during the menstrual cycle in non-pregnant women revealed that, on the one hand, more METs were produced during the luteal phase (LP), especially in comparison to the follicular phase (FP), and on the other hand, that there was a positive relation between the METs amount and progesterone levels in serum samples [77]. Former authors speculated that METs induction might be linked to hormonal changes during the menstruation period with marked LP- and FP-phases [77].

To date, there has been no documentation in scholarly literature regarding the induction of METosis by *C. parvum*. The present findings validate the efficacy of extruded suicidal METs in effectively capturing *C. parvum*-sporozoites and oocysts, thereby potentially hindering sporozoite-mediated host cell invasion and sporozoite excystation through indirect mechanisms. The invasion of host IEC by *C. parvum* sporozoites is a necessary step for their intracellular development into trophozoite-, meront-, and gamont stages. It is plausible to propose that suicidal METosis may serve as a significant defense mechanism in vivo, similar to the reported defense mechanism of suicidal NETosis against ruminant intestinal coccidian parasites such as *E. bovis* and *E. arloingi* [10,65,78]. The discovery regarding oocyst-induced METosis aligns with the observed oocyst-induced NET production in *E. bovis*/*E. arloingi*, which subsequently impedes sporozoite excystation [35]. Similarly, *C. parvum*-oocysts coated by suicidal METs may likewise inhibit sporozoite egress in vivo, thereby interrupting the life cycle at the commencement of endogenous infection.

In order to investigate the dynamic METotic process and to further validate METosis findings in living cells, the experimental setup involved the utilization of a 3D Cell Explorer^®^ microscope in conjunction with an overhead chamber to ensure uniform temperature and CO_2_ ambient conditions. This novel technology permitted the creation of digital stained photographs and three-dimensional renderings of dynamic METotic activities in certain monocytes but also to document resting monocytes, activated cells, and monocytes in close contact with *C. parvum*-sporozoites/oocysts. After *C. parvum* exposure, typical morphological changes in activated monocytes were detected here, as previously reported during parasite-induced NETs [65]. Morphological monocyte changes encompassed the movement of cytoplasmic granules, modification and enlargement of the cell nucleus, cell adherence, uneven surface membranes, and the generation of laminopods/pseudopods aimed to combat parasites.

The intracellular but extracytoplasmic location of *C. parvum* stages (i.e., sporozoites trophozoites, merozoites, gametocytes, and oocysts) places them in a very unique positioning in vivo only separated from intestinal contents by the thin IEC membrane. Moreover, the gut is a distinct niche from other organs with its own physioxia, also described as normoxia, and with it being the largest human organ surface with numerous commensals, micro- and macroorganisms [79,80,81]. Physioxia has been shown to have implications not only in the production of *C. parvum*-triggered NETs [82] but also in the metabolism of bovine small intestinal explants (BSI) and primary bovine intestinal epithelial cells (BSIECs) infected with *C. parvum* [52,53]. Knowing this fact, the metabolic profile of suicidal METosis induced by *C. parvum* was investigated under two different oxygen conditions, namely physioxia (5% O_2_) and hyperoxia (21% O_2_). As already stated, monocytes are known to have significant involvement in the onset and cessation of inflammation, primarily via phagocytosis, the secretion of inflammatory cytokines, reactive oxygen species (ROS), and the stimulation of the adaptive immune system [83]. The aim of this study was to enhance comprehension of distinct metabolic pathways involved in parasite-induced suicidal METosis under intestinal physioxia through the utilization of 5% O_2_ conditions [52,53,79,84]. It should be noted that greater glycolytic activity is typically associated with higher lactate generation and release. Thus, different MCT are important for the transport of lactate across cell membranes, particularly those of monocytes. A total of 14 members make up the MCT family, with MCT1-4 serving as major lactate transporters in both humans and animals, allowing lactate to be absorbed and excreted [81,85,86]. On the other hand, pharmacologic blockers (AR-C 15858 and AR-C 141990), which limit MCT1- and MCT2 activities, respectively, resulted in a non-significant impact on *C. parvum*-induced suicidal METosis with the exception of AR-C 141990 under physioxic condition, which reduced METs formation notably. When it comes to the import of lactate in oxidative cells, MCT1 is frequently implicated, but conversely, MCT4 is more active when it comes to the exportation of lactate resulting from the process of aerobic glycolysis. According to Alarcón et al., it is possible that other transporters besides MCT1/2 may be involved in NETosis induced by parasites [66] and might also occur in *C. parvum*-triggered METosis, but this assumption requests further investigations. MCT1/2 inhibitor treatments, on the other hand, resulted in a reduction in the synthesis of MET formation, especially in physioxic conditions, demonstrating that lactate efflux is definitely important to avoid deleterious cytosol acidification because of increased glycolytic activity induced after stimulation by *C. parvum*. Surprisingly, acidosis caused by increased lactic acid generation in neonatal calves and human children suffering from acute cryptosporidiosis has already been observed in the literature [87,88]. We here show that AR-C 141990 reduced METosis significantly under 5% oxygen condition, showing the possibility of being a good target for drug production.

Some of the most important receptors involved in leukocyte defense mechanisms are the purinergic ATP receptors [89]. As such, purinergic ATP receptors are essential for leukocyte movement, phagocytosis, cell respiration, adhesion, degranulation, and apoptosis [89,90,91,92,93]. Here, we investigated for the first time ATP purinergic receptor P2X1 in *C. parvum*-induced suicidal METosis by using an NF449 inhibitor and demonstrated a P2X1-independent MET process as treatments did not reduce this effector mechanism. According to the current results, parasite infection did not increase the glycolytic responses of bovine monocytes, and treatment with the glycolysis inhibitor 2-DG did not lessen the development of METs caused by *C. parvum*. In contrast to our glycolysis findings in *C. parvum*-exposed monocytes, *B. besnoiti* tachyzoite-triggered bovine NETosis was significantly reduced by FDG treatments [62]. Therefore, it might be linked to cell type differences.

Furthermore, we looked if Notch signaling could be involved in the way apicomplexan parasites cause METosis. Every invertebrate and mammal species share this highly conserved Notch system, which is essential for cellular development processes such as homeostasis, the apoptotic cascade, and cell proliferation. Notch receptors control B- and T-cell maturation in lymphoid organs throughout the body. Nonetheless, the host innate immune system and the creation and activation of myeloid lineages, particularly in the event of acute or chronic inflammation, are not well understood despite Notch signaling implications [49,94]. Monocytes possess the ability to activate Notch signaling pathways through TLR modulation, which is facilitated by the constitutive expression of Notch receptors on their cellular membranes [49,50], and TLR ligands have been shown to activate these cells, which results in the overexpression of ligands such as Jagged-1 or DDL1/4 and Notch receptors [49,50,94,95,96,97,98]. However, there exists a dearth of information regarding Notch-dependent pathways involved in monocytes conducting METosis. The available evidence suggests that Notch signaling plays a significant role in this particular defense mechanism, as evidenced by the fact that respective inhibitor treatments resulted in decreased *C. parvum*-induced MET production under both oxygen conditions, albeit at a marginally higher level of significance. Thus, future monocyte investigations under physioxia conditions will be needed to clarify the precise role of Notch signaling in parasite-triggered METosis. The study presented here has limitations due to the small number of donors and the use of only one *C. parvum* strain. Moreover, it should be noted that all of the experiments in this study were performed under in vitro conditions, implying that the results obtained may not reflect the in vivo intestinal scenario.

## 5. Conclusions

Our findings validated that both stages of *C. parvum* oocysts and sporozoites caused bovine suicidal METosis, which is the natural host of this enteropathogen. Due to the microaerobic replication of *C. parvum* in IEC as well as enteric monocyte recruitment to sites of replication during cryptosporidiosis in vivo [14,16,17], more realistic studies under physioxia are needed in the future. It seems feasible to assume that oocyst trapping through the creation of MET formation will be crucial because it will effectively hamper sporozoite egress, as demonstrated in vivo for closely related *Eimeria* species. Moreover, the lysis of *C. parvum*-infected IEC will expose other parasitic stages (i.e., merozoites, gametocytes, and oocysts) to luminal monocytes, thereby influencing the outcome of infection. The present study effectively addressed the benefits and utility of live cell 3D holotomographic microscopy imaging in investigating the METotic process induced by *C. parvum*. Similarly, Seahorse^®^-based glycolytic measurements have been instrumental in enhancing comprehension of metabolic signatures of monocytes during MET production. Although current in vitro results clearly show *C. parvum*-induced suicidal METosis, further studies are required to be performed in vivo to confirm the pivotal function of this innate immune mechanism in bovine and human cryptosporidiosis.

## Figures and Tables

**Figure 1 biology-12-00961-f001:**
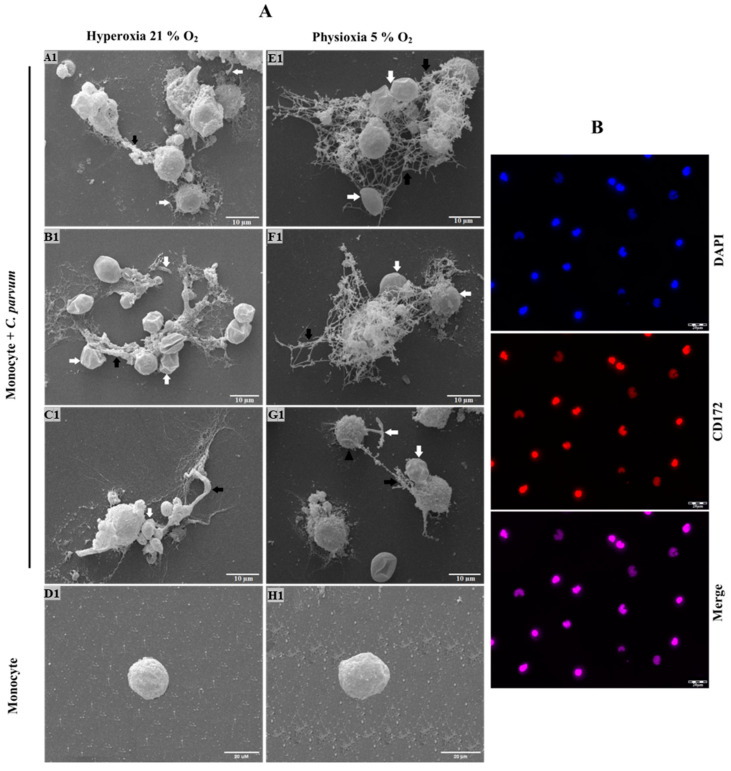
SEM analysis was utilized to evaluate the *Cryptosporidium parvum*-induced suicide METosis (*n* = 3) under physioxic (5% O_2_) and hyperoxic (21% O_2_) circumstances. (**A**) Oocysts of *C. parvum* were exposed to isolated bovine monocytes and thereafter fixed for SEM analysis. The presence of filaroid structures resembling METs, characterized by their delicate and thick nature, was revealed, as indicated by the black arrows. The occurrence of extruded suicidal METosis is characterized by the observation of tightly entrapped oocysts and sporozoites (indicated by white arrows) within ruptured monocytes. Moreover, inactivated monocytes (**D1**,**H1**) with rough surfaces and activated cells (**G1**, black arrow heads) were also detected in these reactions but without complete cell disruption. (**B**) The CD-172+ marker was used for monocyte confirmation.

**Figure 2 biology-12-00961-f002:**
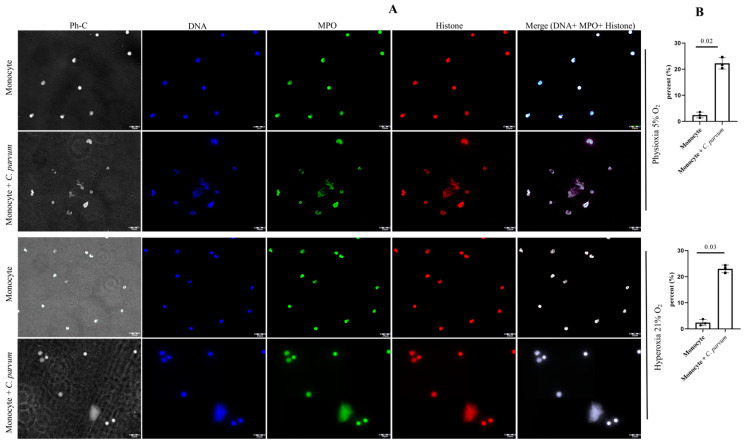
(**A**) The METosis phenomenon (*n* = 3), induced by *Cryptosporidium parvum*, is distinguished by the convergence of monocyte-derived DNA (blue) embellished with myeloperoxidase (MPO, green) and global histones (H1, H2A/B, H3, H4, red), leading to the amalgamation of the aforementioned three channels. (**B**) METs percentage is shown as well for both oxygen concentrations.

**Figure 3 biology-12-00961-f003:**
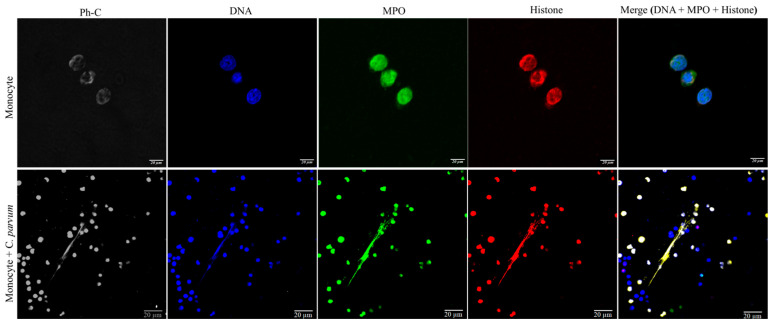
*Cryptosporidium parvum*-triggered suicidal METosis (*n* = 3) is demonstrated using confocal microscopy by co-localization of monocyte-derived DNA (blue) decorated with myeloperoxidase (MPO, green), histones (H1, H2A/B, H3, H4, red), and the resulting merging of the three channels.

**Figure 4 biology-12-00961-f004:**
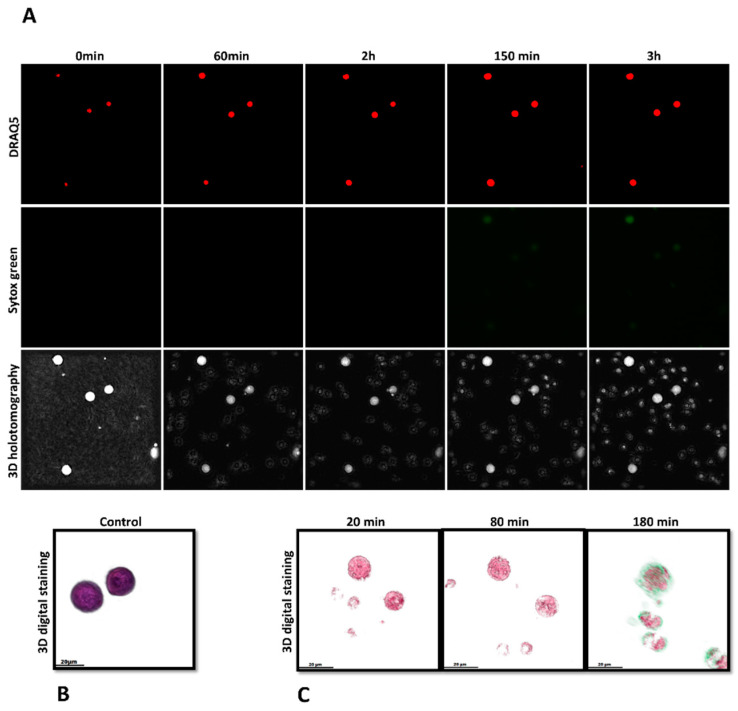
Investigating live-cell 3D-holotomographic microscopy to observe the interactions between monocytes and *C. parvum* (sporozoites/oocysts) in a three-dimensional environment. (**A**) depicts a time-lapse visualization of monocytes that have been subjected to staining with SYTOX Green^®^ and DRAQ5^®^. (**B**) demonstrates the utilization of 3D-digital staining technique for the visualization of control experiment involving inactivated monocytes. (**C**) The study utilized three-dimensional digital staining to illustrate the morphological changes that monocytes exposed to *C. parvum* underwent at 20, 80, and 180 min of stimulation, respectively.

**Figure 5 biology-12-00961-f005:**
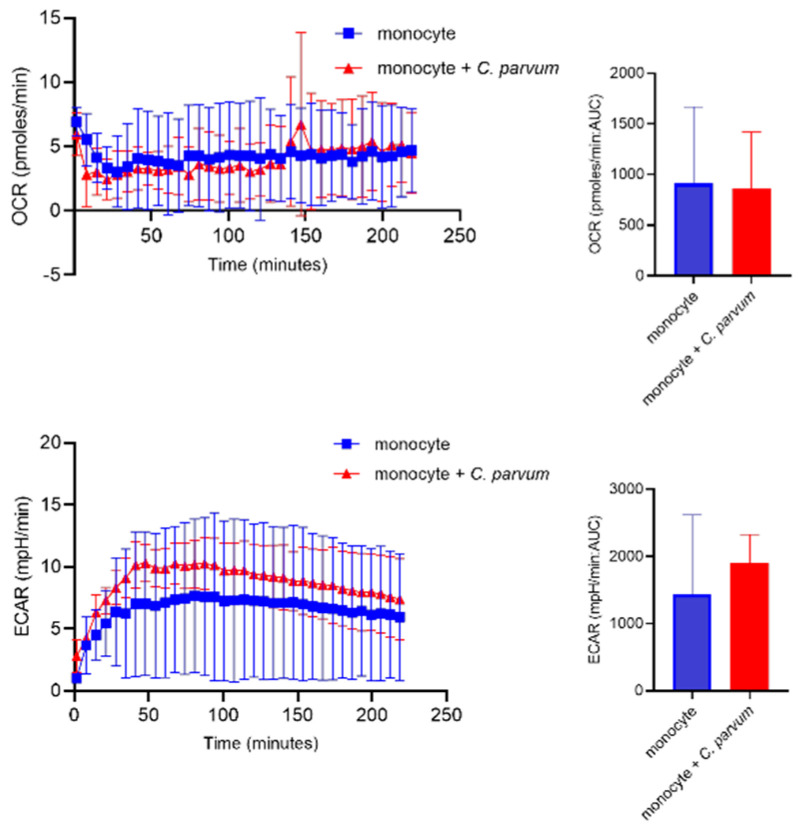
Assessment of the metabolic changes that occur in bovine monocytes when exposed to *Cryptosporidium parvum*. Evaluation of oxygen consumption rate (OCR) and extracellular acidification rate (ECAR) in bovine monocytes using an extracellular flux analyzer (Seahorse^®^; Agilent) following injection of *C. parvum* into the plate. Measured OCR- and ECAR concentrations during 220 min. The area under the curve (AUC) was calculated for all registries and given as mean ± SD.

**Figure 6 biology-12-00961-f006:**
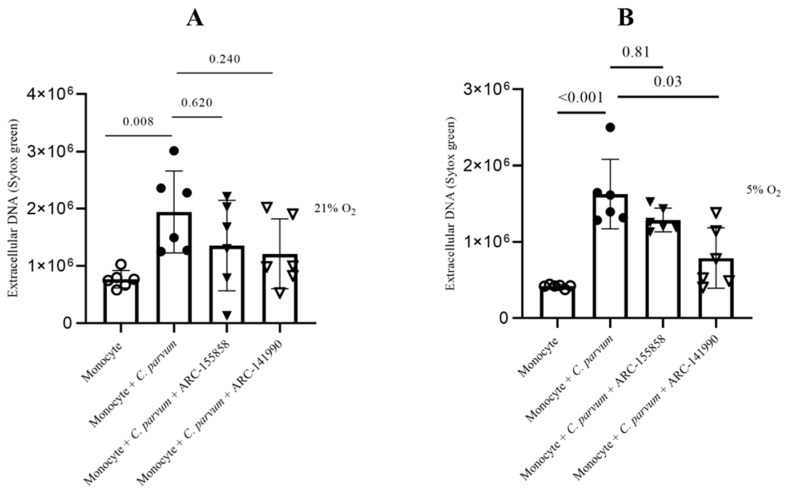
The impact of treatments with AR-C 155858 and AR-C 141990 on suicidal METosis (*n* = 6) induced by *Cryptosporidium parvum*. Hyperoxic (**A**) and physioxic (**B**) co-culture conditions. Bovine monocytes were subjected to a pre-treatment of AR-C 155858 (1 μM) and AR-C 141990 (1 μM) for a duration of 30 min. Following this, during a 3 h time frame, monocytes were subjected to *C. parvum*. The assessment of extracellular DNA was conducted by measuring fluorescence intensities derived from SYTOX Green^®^ using multi-plate reader (Varioskan^®^, Thermo Scientific). The graphs present the mean value along with the standard deviation (SD), and statistical significance was determined by a *p*-value less than 0.05.

**Figure 7 biology-12-00961-f007:**
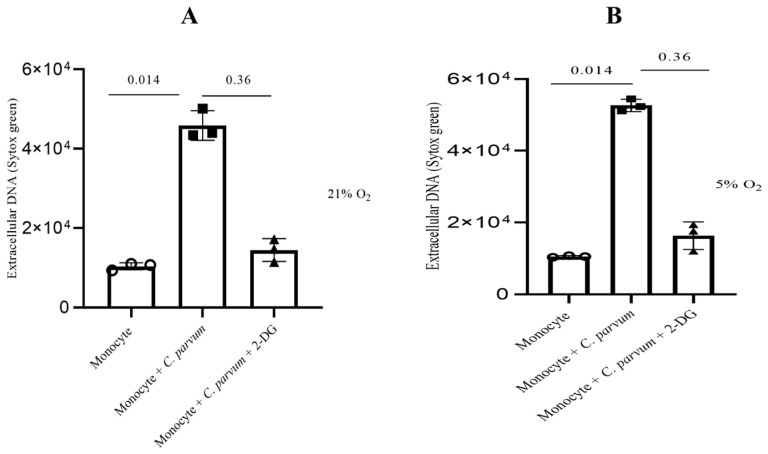
The impact of glycolysis under hyperoxic (**A**) and physioxic (**B**) conditions during *Cryptosporidium parvum*-induced METosis. Before exposing bovine monocytes (*n* = 3) to *C. parvum* (in a proportion of 1:3) for three hours, they were given a pre-treatment of 2 mM for 2-DG (30 min), which is an inhibitor of glycolysis. The fluorescence intensities of SYTOX Green^®^ were measured using an automated multi-plate reader (Varioskan^®^, Thermo Scientific) to evaluate extracellular DNA. Results show mean ± SD and *p* values less than 0.05 as statistically significant.

**Figure 8 biology-12-00961-f008:**
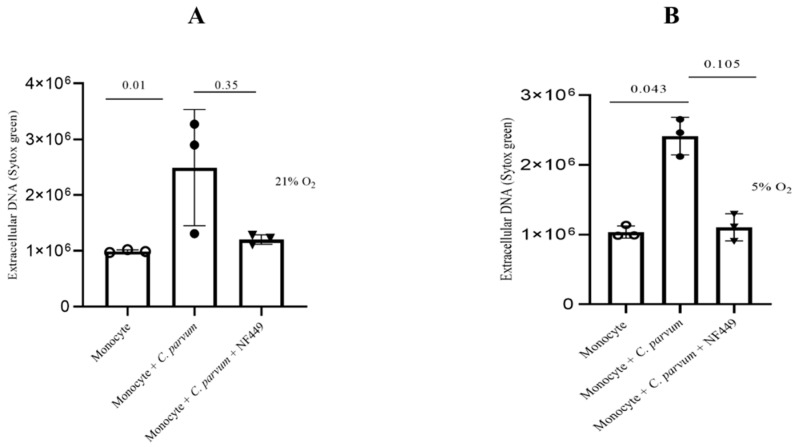
The effects of hyperoxia (**A**) and physioxia (**B**) on P2X1-mediated purinergic signaling in *C. parvum*-induced METosis. Bovine monocytes (*n* = 3) were subjected to pre-treatment with NF449 (100 μM), a P2X1 inhibitor, for a duration of 30 min prior to being exposed to *C. parvum* (proportion 1:3) for a period of 3 h. The fluorescence intensities of SYTOX Green^®^ were measured using an automated multi-plate reader (Varioskan^®^, Thermo Scientific) to evaluate extracellular DNA. The reported results in the graphs consist of the mean ± SD of the mean. Statistical significance was determined by *p*-values less than 0.05.

**Figure 9 biology-12-00961-f009:**
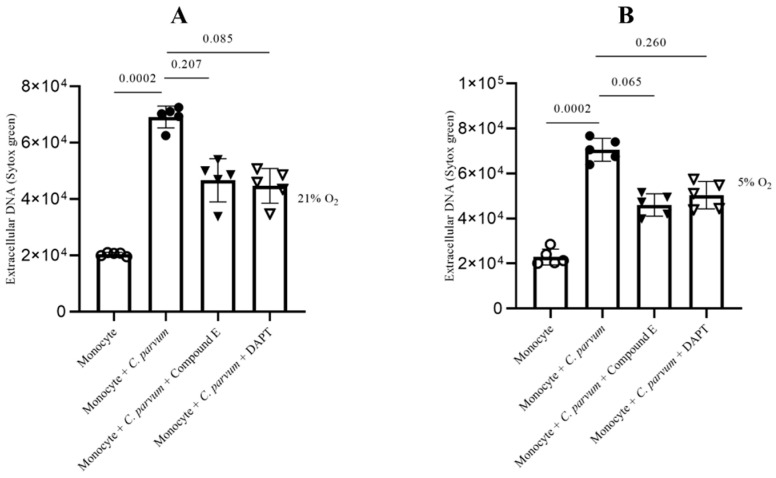
The impact of Notch inhibitors on suicidal METosis induced by *Cryptosporidium parvum* under hyperoxia (21% O_2_; (**A**)) and physioxia (5% O_2_; (**B**)) is being investigated. Prior to exposure to *C. parvum* (proportion 1:3), bovine monocytes (*n* = 5) were subjected to pre-treatment with 50 μM of DAPT, an inhibitor of Notch signaling, and 20 nM of compound E, another inhibitor of Notch signaling, for a duration of 30 min. The fluorescence intensities of SYTOX Green^®^ were measured using an automated multi-plate reader (Varioskan^®^, Thermo Scientific) to evaluate extracellular DNA. The reported results in the graphs consist of the mean ± SD. Statistical significance was determined by considering *p*-values less than 0.05.

## Data Availability

The present study incorporates all data produced within its scope in this article and its Appendix A.

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
