# Peer review of "MCT-Dependent Cryptosporidium parvum-Induced Bovine Monocyte Extracellular Traps (METs) under Physioxia"

_biology, 2023, doi:10.3390/biology12070961_

Round 1
Reviewer 1 Report
This paper describes the induction of METosis after exposing bovine monocytes to C. parvum sporozoites and oocysts. The authors then attempt to determine the pathways that might be involved in this defense mechanism. Monocarboxylate transporters were shown to play a role in METosis. A similar paper published last year by the same group, (which might be worth mentioning in the introduction) examined neutrophiles (NETosis) after exposure to C. parvum sporozoites and ATP inhibitors. The present study is novel in that METs have not been looked at against C. parvum. Methods for the most part are well described and the paper is well written. As pointed out by the authors it is too early to determine the relevance of this defense mechanism since this process has not been shown to occur in vivo with C. parvum and most of the parasite’s life cycle are intracellular. None the less, it is the first identification of a potential innate immune mechanism of this type against C. parvum.
Comments:
In Figure 2, it states that exposure to sporozoites increased METosis and percentage of METs was higher. Please clarify how Mets were quantitated. There seems to be both individual and cluster of METs. Were individual monocytes counted or clusters of cells.
CD127 is used to identify monocyte populations (Figure 1). Cell are nicely labelled. However, it is not explained how pure the cell population is after the isolation preparation. Is there a possibility that other cells could be contained in the preparation that were not labelled and contributed to METosis? Flow cytometry could be used to verify the purity of the cell populations.
Monocytes cells are derived from PBMCs, how relevant are these cells to monocytes found in the gut?
Figure 7. Graphs looks like it should be statically significant (large reduction on Mets with non overlapping SD bars) but apparently was not. Is this due to low sample number? Might be worth mentioning why this is the case and if additional experiments yielded the same results (clear reductions with the inhibitors but statistically significantly).
All captions should state sample size (n number) and also number of repetitions performed for each experiment.
Author Response
Thank you for your comments and please find the reply attachment

Reviewer 2 Report
The manuscript entitled ' Cryptosporidium parvum-induced bovine monocyte extracellu- 2 lar traps (METs) MCT-dependant under physioxia' by Hasheminasab et al., describes In C. parvum-exposed bovine monocytes, the role of ATP purinergic receptor P2X1, glycolysis, monocarboxylate transporters (MCT), and Notch signalling was studied in terms of METs under intestine physioxia and hyperoxia. The manuscript is well written, below are some suggestions that authors should address.
1. In the monocytes were culture in RPIM media without FBS, generally most of the report suggests that monocytes culture should be done with 10% FBS with RPIM or DMEM, I am curious was there any reason why only RPIM was used for the culture.
2. Did the authors face contamination issues while the culture of monocytes?
3. The mammalian-derived extracellular traps (METs) in stimulated monocytes induced by C. parvum were confirmed through immunofluorescence against different histones proteins. What about a quantitative assay of proteins such as western blot? Why authors did not perform immunoblot. Immunofluorescence can only detect the signal. Figure 2 and 3.
Authors should combine a Simple Summary and abstract so this will be easy for readers to understand the study.
I recommended the publication of this article after revision.
English is fine
Author Response

(The authors gave the same response as above.)

Reviewer 3 Report
The manuscript "Cryptosporidium parvum-induced bovine monocyte extracellular traps (METs) MCT-dependant under physioxia" is an interesting study, which can provide insights for controlling cryptosporidiosis. However, there are numerous type problems not limited to the following issues in the whole MS. I think the MS is not fit for the quality of Biology at current condition.
1. The title is confusing, and need to be improved.
2. “Cryptosporidium parvum” needs to be italic.
3. The author compared two groups (intestinal physioxia and hyperoxia), why not include normal or physiological condition.
4. Lin56: “5% CO2” should be “5% O2”
5. Lin137: “IIaA15G2RI” should be “IIaA15G2R1”, the results for the identification of species and subtypes should be included in the MS or as a supplemental file.
6. There are many format problems, such as, sometimes used “degrees”, but sometimes for “°C”;
7. As to Figure 2, Merge group represents which two groups? And the figure lacks of results under normal light. The same as Figure 3.
8. The scale bars in figures 2 and 3 were not clear.
9. As to Figure 5, no significant differences were likely caused by high error bars.
10. In discussion, paragraphs 2 and 3 need to be merged in one paragraph.
11. Limitations of the present study need to be addressed at the end of discussion.
Moderate editing of English language required
Author Response

(The authors gave the same response as above.)

Round 2
Reviewer 2 Report
The authors have addressed the comments,
the manuscript should be accepted for the publication.
minor corrections required
Reviewer 3 Report
The authors have addressed most problems, and I have no further problems.
Minor editing of English language required